# A lectin receptor kinase as a potential sensor for extracellular nicotinamide adenine dinucleotide in *Arabidopsis thaliana*

Chenggang Wang[1†], Mingqi Zhou[1†], Xudong Zhang[1], Jin Yao[2], Yanping Zhang[3], Zhonglin Mou[1*]

[1]Department of Microbiology and Cell Science, University of Florida, Gainesville, United States; [2]Target Sciences, GlaxoSmithKline, King of Prussia, United States; [3]Interdisciplinary Center for Biotechnology Research, University of Florida, Gainesville, United States

*For correspondence: zhlmou@ufl.edu

[†]These authors contributed equally to this work

Competing interests: The authors declare that no competing interests exist.

**Abstract** Nicotinamide adenine dinucleotide ($NAD^+$) participates in intracellular and extracellular signaling events unrelated to metabolism. In animals, purinergic receptors are required for extracellular $NAD^+$ ($eNAD^+$) to evoke biological responses, indicating that $eNAD^+$ may be sensed by cell-surface receptors. However, the identity of $eNAD^+$-binding receptors still remains elusive. Here, we identify a lectin receptor kinase (LecRK), LecRK-I.8, as a potential $eNAD^+$ receptor in *Arabidopsis*. The extracellular lectin domain of LecRK-I.8 binds $NAD^+$ with a dissociation constant of $436.5 \pm 104.8$ nM, although much higher concentrations are needed to trigger in vivo responses. Mutations in *LecRK-I.8* inhibit $NAD^+$-induced immune responses, whereas overexpression of *LecRK-I.8* enhances the *Arabidopsis* response to $NAD^+$. Furthermore, *LecRK-I.8* is required for basal resistance against bacterial pathogens, substantiating a role for $eNAD^+$ in plant immunity. Our results demonstrate that lectin receptors can potentially function as $eNAD^+$-binding receptors and provide direct evidence for $eNAD^+$ being an endogenous signaling molecule in plants.

## Introduction

The pyridine nucleotide $NAD^+$ (nicotinamide adenine dinucleotide) not only serves as a ubiquitous coenzymatic redox carrier in metabolic reactions, but also participates in intracellular signal transduction (*Berger et al., 2004*; *Noctor et al., 2006*). $NAD^+$ is the precursor of the second messenger cyclic ADP-ribose (cADPR), which triggers calcium ($Ca^{2+}$) release from intracellular stores in various organisms (*Galione and Churchill, 2000*; *Lee, 2001*; *Hunt et al., 2004*; *Ziegler, 2005*). $NAD^+$ also functions as the ADP-ribose donor and the acetyl group acceptor in protein ADP-ribosylation and deacetylation reactions, respectively (*Jacobson and Jacobson, 1999*; *Bürkle, 2001*; *Denu, 2003*; *Jackson et al., 2003*; *Hunt et al., 2004*). In response to environmental stimuli, cellular $NAD^+$ can also be released into the extracellular space by active exocytosis or diffusion through transmembrane transporters in living cells or passive leakage across membrane in collapsed tissues (*Bruzzone et al., 2001*; *Contreras et al., 2003*; *Seman et al., 2003*; *Adriouch et al., 2007*). It has recently been shown that extracellular $NAD^+$ ($eNAD^+$) plays an important signaling role in numerous physiological and pathological processes (*Billington et al., 2006*; *Haag et al., 2007*; *Adriouch et al., 2012*).

In animal cells, $eNAD^+$ can be processed by ectoenzymes such as CD38, CD157, and mono(ADP-ribosyl)transferases (ARTs) (*Billington et al., 2006*). CD38 is a multifunctional enzyme attached to

**eLife digest** Plants and animals are generally healthy, despite being surrounded by many different microbes that have the potential to infect them and cause disease. This is because plants and animals are able to sense infections and promptly activate immune responses against them. A molecule known as NAD is involved in many processes inside healthy cells, but it can also act as a warning signal of infection. When an invading microbe damages a host cell, NAD leaks out of the damaged cell. Neighboring healthy cells sense this NAD and activate immune responses.

It is thought that specific receptor proteins on the surface of animal and plant cells are responsible for detecting NAD that has leaked out of damaged cells. However, the identities of these receptors were not known. Wang, Zhou et al. used genetics and biochemical techniques to investigate how cells in a plant known as *Arabidopsis* detect NAD.

The experiments reveal that a receptor protein called LecRK-I.8 can bind NAD via a section of the receptor known as the lectin domain. *Arabidopsis* plants with mutant forms of LecRK-I.8 are less able to activate immune responses when exposed to NAD compared to normal plants. Furthermore, the mutant plants are less able to defend themselves against *Pseudomonas syringae*, a bacterium that can infect many different plants. On the other hand, plants with higher levels of LecRK-I.8 than normal produce stronger immune responses to NAD.

The findings of Wang, Zhou et al. identify the first receptor on the surface of plant cells that can detect NAD. The next challenge is to find out if humans and other animals also use similar proteins to detect NAD during infections. In agriculture, bacterial infections can lead to major losses of crops. Therefore, these findings may help researchers to develop crop varieties that are more resistant to these infections.

the extracellular surface of the plasma membrane, which utilizes $NAD^+$ as the substrate to produce cADPR (*Ceni et al., 2003*; *De Flora et al., 2004*; *Krebs et al., 2005*; *Malavasi et al., 2006*; *Morabito et al., 2006*; *Partidá-Sánchez et al., 2007*). ARTs are glycosylphoshpatidylinositol-anchored or secreted ectoenzymes that use $NAD^+$ to ADP-ribosylate lipid raft-associated signaling proteins (*Nemoto et al., 1996*; *Han et al., 2000*; *Seman et al., 2003*; *Bannas et al., 2005*; *Zolkiewska, 2005*). $eNAD^+$ may also be perceived by cell-surface receptors. *Moreschi et al. (2006)* reported that $NAD^+$ promotes intracellular $[Ca^{2+}]$ elevation in the human purinoceptor $P2Y_{11}$-transfected astrocytoma cells, but not in untransfected cells. They also showed that, in human granulocytes, treatment with the selective and potent $P2Y_{11}$ inhibitor NF157 and down-regulation of $P2Y_{11}$ expression by short interference RNA, both prevented $NAD^+$-induced intracellular $[Ca^{2+}]$ increases and chemotaxis (*Moreschi et al., 2006*). These results demonstrate that $P2Y_{11}$ is involved in $eNAD^+$-triggered transmembrane signaling. Several other studies using similar approaches have also indicated that purinoceptors, including $P2Y_1$, $P2X_1$, $P2X_4$, and $P2X_7$, are engaged in $eNAD^+$-mediated signaling (*Mutafova-Yambolieva et al., 2007*; *Grahnert et al., 2009*; *Klein et al., 2009*). However, $NAD^+$ per se has never been demonstrated to bind to purinoceptors, and thus the identity of $eNAD^+$ receptors still remains a mystery in animals.

In plants, $NAD^+$ and its derivatives have also been shown to function in stress tolerance and/or defense signaling (*Dutilleul et al., 2005*; *Adams-Phillips et al., 2010*; *PetriacqPétriacq et al., 2013*). Overexpression of the bacterial $NAD^+$ biosynthesis gene *nadC*, which increases intracellular $NAD^+$ levels, enhances defense gene expression and bacterial pathogen resistance (*Pétriacq et al., 2012*). In contrast, mutations in *FLAGELLIN-INSENSITIVE4*, a de novo $NAD^+$ biosynthesis gene, suppress stomatal immunity (*Macho et al., 2012*). Furthermore, overexpression of *Arabidopsis thaliana Nudix hydrolase homolog 6* (*AtNUDT6*), which encodes an ADP-ribose/NADH pyrophosphohydrolase, and knockout of *AtNUDT6*, *AtNUDT7*, or *AtNUDT8* lead to alterations of intracellular NADH levels and salicylic acid (SA)-mediated defense signaling (*Bartsch et al., 2006*; *Ge et al., 2007*; *Ishikawa et al., 2010*; *Fonseca and Dong, 2014*).

We have recently shown that exogenous $NAD^+$ induces SA-dependent and -independent *PATHOGENESIS-RELATED* (*PR*) gene expression and disease resistance (*Zhang and Mou, 2009*). Importantly, we found that pathogen-induced hypersensitive response causes leakage of $NAD^+$ into the

extracellular fluid at concentrations sufficient to induce *PR* gene expression and disease resistance (*Zhang and Mou, 2009*). These results provided the first line of evidence that $NAD^+$ may also play a signaling role in plant extracellular space. However, since proteins with significant homology to animal CD38/CD157, ARTs, and purinoceptors are absent in plants, it remains unclear if $eNAD^+$ is an endogenous signaling molecule in plants and if plants use similar mechanisms to process or perceive $eNAD^+$ (*Hunt et al., 2004*; *Sánchez et al., 2004*; *Zolkiewska, 2005*; *Zhang and Mou, 2008*). We have shown that expression of the human $NAD^+$-metabolizing ectoenzyme CD38 partially compromises systemic acquired resistance (SAR) in *Arabidopsis* (*Zhang and Mou, 2012*), which strongly suggests that plants may use different mechanisms to sense $eNAD^+$.

In order to understand $eNAD^+$ and its signaling role in plants, we performed a forward genetic screen in *Arabidopsis* to identify mutants insensitive to exogenous $NAD^+$ (*ien*) treatment (*Zhang et al., 2012*). Characterization of several *ien* mutants revealed that the Mediator complex subunits MED14/STRUWWELPETER and MED16/SENSITIVE TO FREEZING6 /IEN1 as well as the Elongator complex function downstream of $eNAD^+$ (*Zhang et al., 2012*, *2013*; *An et al., 2016*). However, no receptor genes were identified in the forward genetic screen. In this study, we employed a reverse genetic approach to identify $eNAD^+$ receptors in *Arabidopsis*. We demonstrate that the lectin receptor kinase (LecRK), LecRK-I.8, is a potential $eNAD^+$ receptor and plays a positive role in plant basal immunity. Our findings indicate that cell-surface lectin receptors can potentially act as $eNAD^+$-sensing receptors and present direct evidence for $eNAD^+$ being a novel endogenous signaling molecule in plants.

## Results

### Exogenous $NAD^+$ induces profound transcriptome changes toward immune response

Thus far, we have only shown that exogenous application of $NAD^+$ induces *PR* gene expression in the model plant *Arabidopsis* (*Zhang and Mou, 2009*). To identify genes induced by $NAD^+$ at the genome level, we performed a microarray experiment to monitor $NAD^+$-induced transcriptome changes in wild-type Col-0 plants (National Center for Biotechnology Information Gene Expression Omnibus series number GSE76568). Triplicate experiments were performed independently, and the data were analyzed to identify genes that showed a twofold or higher induction or suppression with a low *q* value ($\leq$0.05). Compared to the mock (water) treatment, $NAD^+$ addition caused profound transcriptional changes, including upregulation of 2155 genes and downregulation of 2014 genes. In the upregulated genes, those involved in plant immune responses were significantly enriched, whereas in the downregulated genes, those associated with responses to hormone stimuli, such as auxin stimulus, were overrepresented (*Figure 1A*). The $NAD^+$-upregulated genes include a large number of pathogen-associated molecular pattern (PAMP)-triggered immunity (PTI) and SA pathway genes (*Supplementary file 1A*). In contrast, expression of several jasmonic acid (JA)/ethylene (ET)-mediated defense pathway genes, including the widely used defense marker gene *PLANT DEFEN-SIN1.2*, was suppressed by $NAD^+$ treatment (*Supplementary file 1A*). Interestingly, more than 88% of the $NAD^+$-induced genes were also activated by the bacterial pathogen *Pseudomonas syringae* pv. *tomato* (*Pst*) DC3000 carrying the effector avrRpt2 (*Figure 1B*) (*Wang et al., 2013*). These results are in agreement with the strong resistance induced by $NAD^+$ against the hemibiotrophic bacterial pathogen *P. syringae* (*Zhang and Mou, 2009*).

### Mutations in the *LecRK-I.8* gene inhibit $NAD^+$-induced *PR* gene expression and disease resistance

Further analysis of the microarray data revealed that a group of receptor kinase (RK) including several cell wall-associated kinase (WAK) genes were induced by the $NAD^+$ application (*Supplementary file 1B*). We isolated transferred DNA (T-DNA) insertion lines for fourteen of the RK genes (*Supplementary file 1B*), tested their responsiveness to $NAD^+$, and found that $NAD^+$-induced resistance to the bacterial pathogen *P. syringae* pv. *maculicola* (*Psm*) ES4326 was reduced in one T-DNA insertion line (Arabidopsis Biological Resource Center accession code Salk_066416) (*Figure 2—figure supplement 1A*). Salk_066416 carries a T-DNA insertion in the gene At5g60280, which was predicted to encode the legume-like (L-type) lectin receptor kinase-I.8 (LecRK-I.8)

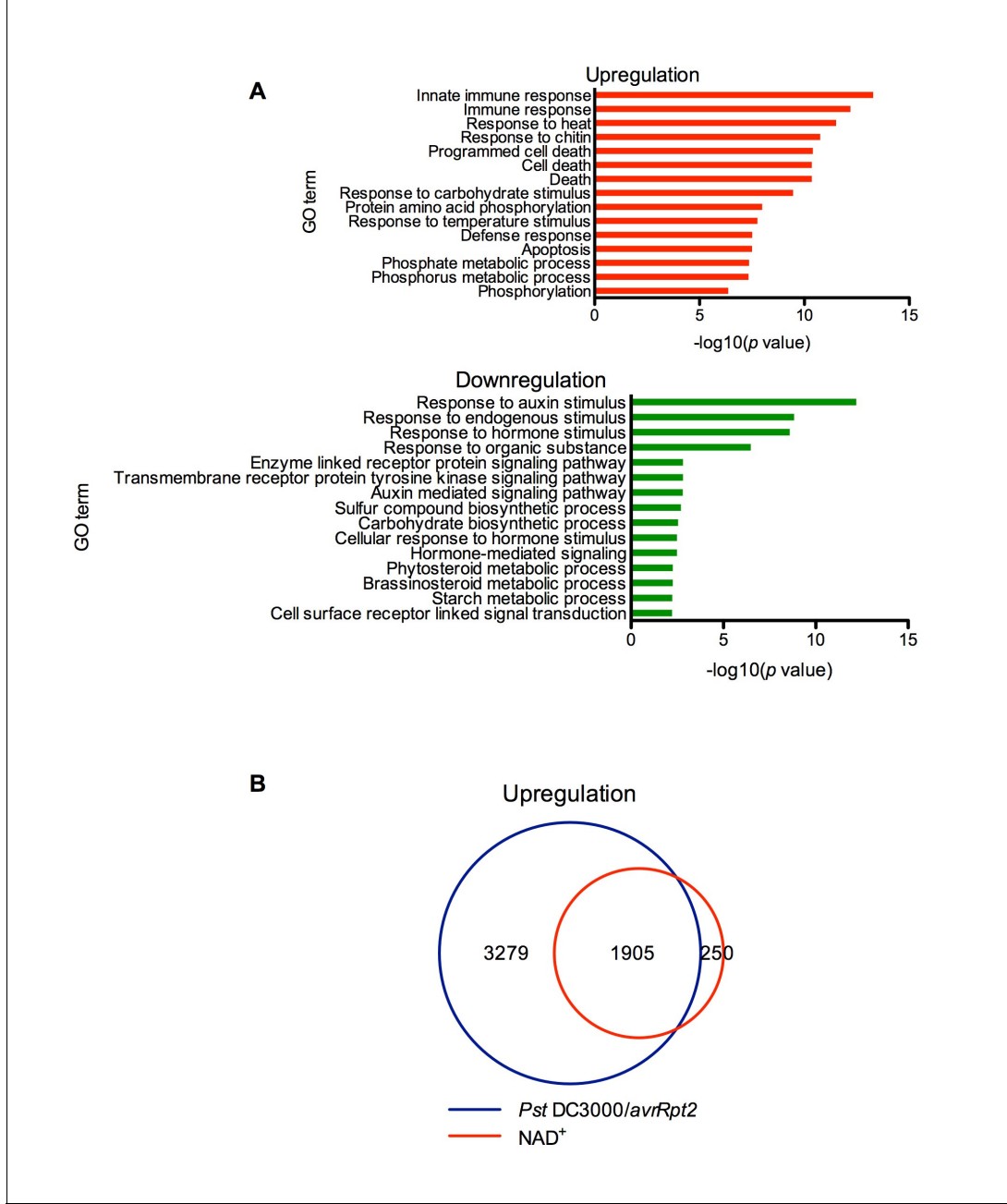

**Figure 1.** Exogenous NAD$^+$-induced transcriptome changes. (**A**) Gene Ontology (GO) term enrichment test of the genes that were upregulated and downregulated by NAD$^+$ treatment at 4 hr showed that genes involved in plant defense such as innate immune response, immune response, and response to chitin were significantly enriched in the upregulated genes, whereas those associated with responses to hormone stimuli, such as auxin stimulus, were overrepresented in the downregulated genes. (**B**) Overlap between the genes that were upregulated by NAD$^+$ treatment at 4 hr and that by *Pst* DC3000/*avrRpt2* at least at one time point of 4, 8, and 12 hr post-inoculation (**Wang et al., 2013**).

(**Bouwmeester and Govers, 2009**). We then isolated two more T-DNA insertion lines (Salk_005125 and Salk_206382) for further investigation. The three T-DNA insertion lines had reduced transcript levels (**Figure 2—figure supplement 1B and C**). Salk_066416 has previously been named *lecrk-I.8–2* (**Wang et al., 2014**). Salk_005125 and Salk_206382 were thus named *lecrk-I.8–3*, and *lecrk-I.8–4*, respectively (**Figure 2—figure supplement 1B**). The *lecrk-I.8–4* mutant accumulated higher transcript levels than *lecrk-I.8–2* and *lecrk-I.8–3*, and was considered a weak allele (**Figure 2—figure supplement 1C**). NAD$^+$-induced expression of *PR1* and resistance to *Psm* ES4326 were significantly

inhibited in all three *lecrk-I.8* alleles but not in the extracellular ATP (eATP) receptor mutant *dorn1-3* (*Choi et al., 2014*), and induction of *PR2* and *PR5* was also repressed in *lecrk-I.8–2* and *lecrk-I.8–3* (*Figure 2A–D*). These results indicate that LecRK-I.8 is a component of the eNAD$^+$-induced defense signaling pathway. Interestingly, induction of *PR2* and *PR5* was significantly enhanced in the *dorn1-3* mutant (*Figure 2B and C*), which is consistent with eATP being a negative regulator of SA signaling (*Chivasa et al., 2009*).

To test if overexpression of *LecRK-I.8* could enhance *Arabidopsis* responsiveness to NAD$^+$, we transformed a *35S:LecRK-I.8* construct into wild-type Col-0 plants. Intriguingly, we were unable to identify transgenic lines expressing very high levels of *LecRK-I.8*. The expression levels of *LecRK-I.8* in all of the obtained 11 transgenic lines increased less than fourfold (*Figure 2—figure supplement 2A*), suggesting that overexpression of *LecRK-I.8* may be detrimental to plant growth and development. Nevertheless, although the transgenic lines, which expressed higher levels of *LecRK-I.8* than the wild type, did not show enhanced disease resistance (*Figure 2—figure supplement 2B*), NAD$^+$-induced expression of *PR1* and *PR5* in these lines was significantly enhanced (*Figure 2E–2G*). This result supports that LecRK-I.8 functions in eNAD$^+$-mediated defense pathway.

## LecRK-I.8 is located in the plasma membrane

Both DORN1 and LecRK-I.8 contain a putative transmembrane domain and two putative arginine-glycine-aspartic acid (RGD)-binding motifs that likely mediate plant cell wall-plasma membrane interactions (*Gouget et al., 2006*) (*Figure 3A* and *Figure 3—figure supplement 1*), suggesting their possible plasma membrane localization. Indeed, DORN1 has been shown to be localized in the plasma membrane (*Bouwmeester et al., 2011*). To determine the subcellular localization of LecRK-I.8, we transformed a *35S:LecRK-I.8-GFP* (*Green Fluorescence Protein*) construct into the *lecrk-I.8–2* mutant. Very low levels of LecRK-I.8-GFP protein were detected in three out of eleven single insertion homozygous transgenic lines (*Figure 3—figure supplement 2A*), which is consistent with the hypothesis that overexpression of *LecRK-I.8* may be harmful to plants. Nevertheless, the low level of LecRK-I.8-GFP complemented the enhanced disease susceptibility phenotype of *lecrk-I.8–2* (see below) (*Figure 3—figure supplement 2B*), indicating that LecRK-I.8-GFP is biologically active. Unfortunately, we were unable to detect any GFP fluorescence in the transgenic *Arabidopsis* plants accumulating LecRK-I.8-GFP. To circumvent this problem, we transiently expressed LecRK-I.8-GFP in *Nicotiana benthamiana*. The LecRK-I.8-GFP fusion protein in *N. benthamiana* appeared to be localized in the plasma membrane (*Figure 3B*). To confirm this subcellular localization, we co-expressed LecRK-I.8-GFP and AHA2-mCherry in *N. benthamiana*. As shown in *Figure 3C*, the GFP and mCherry signals in the co-transformed *N. benthamiana* epidermal cell precisely overlapped with each other. Since *AHA2* encodes a well-established plasma membrane localized P-type H$^+$- ATPase (*DeWitt et al., 1996*), this result indicates that LecRK-I.8 is localized at the plasma membrane.

## The kinase domain of LecRK-I.8 possesses kinase activity

LecRK-I.8 contains a cytoplasmic kinase domain (KD) (*Figure 3—figure supplement 1*). To test if LecRK-I.8 is an active kinase, we expressed the LecRK-I.8KD fragment as a Maltose-Binding Protein (MBP)-LecRK-I.8KD fusion protein in *Escherichia coli*. The MBP-LecRK-I.8KD recombinant protein was purified using amylose resin and subjected to kinase activity assay (*Figure 4*). Simultaneously purified MBP protein was included in the experiment as a negative control (*Figure 4*). As shown in the autoradiograph, the LecRK-I.8KD protein exhibited a strong autophosphorylation activity and also phosphorylated myelin basic protein, a substrate commonly used for in vitro kinase assays. This result indicates that LecRK-I.8 possesses kinase activity.

## LecRK-I.8 specifically binds NAD$^+$

Plasma membrane localization of LecRK-I.8 plus its active cytoplasmic kinase domain suggest a putative role for LecRK-I.8 as a receptor for an extracellular ligand. To test whether NAD$^+$ is a ligand binding to LecRK-I.8, we generated transgenic *Arabidopsis* plants expressing the extracellular lectin domain (amino acids (AAs) 23 to 283) of LecRK-I.8 fused to GFP (eLecRK-I.8-GFP). The eLecRK-I.8-GFP protein was immunoprecipitated using an anti-GFP antibody (*Figure 5—figure supplement 1A*) and subjected to binding assays with $^{32}$P-labeled NAD$^+$. A significant NAD$^+$ binding activity was detected for the immunoprecipitated eLecRK-I.8-GFP protein, but not for the GFP protein

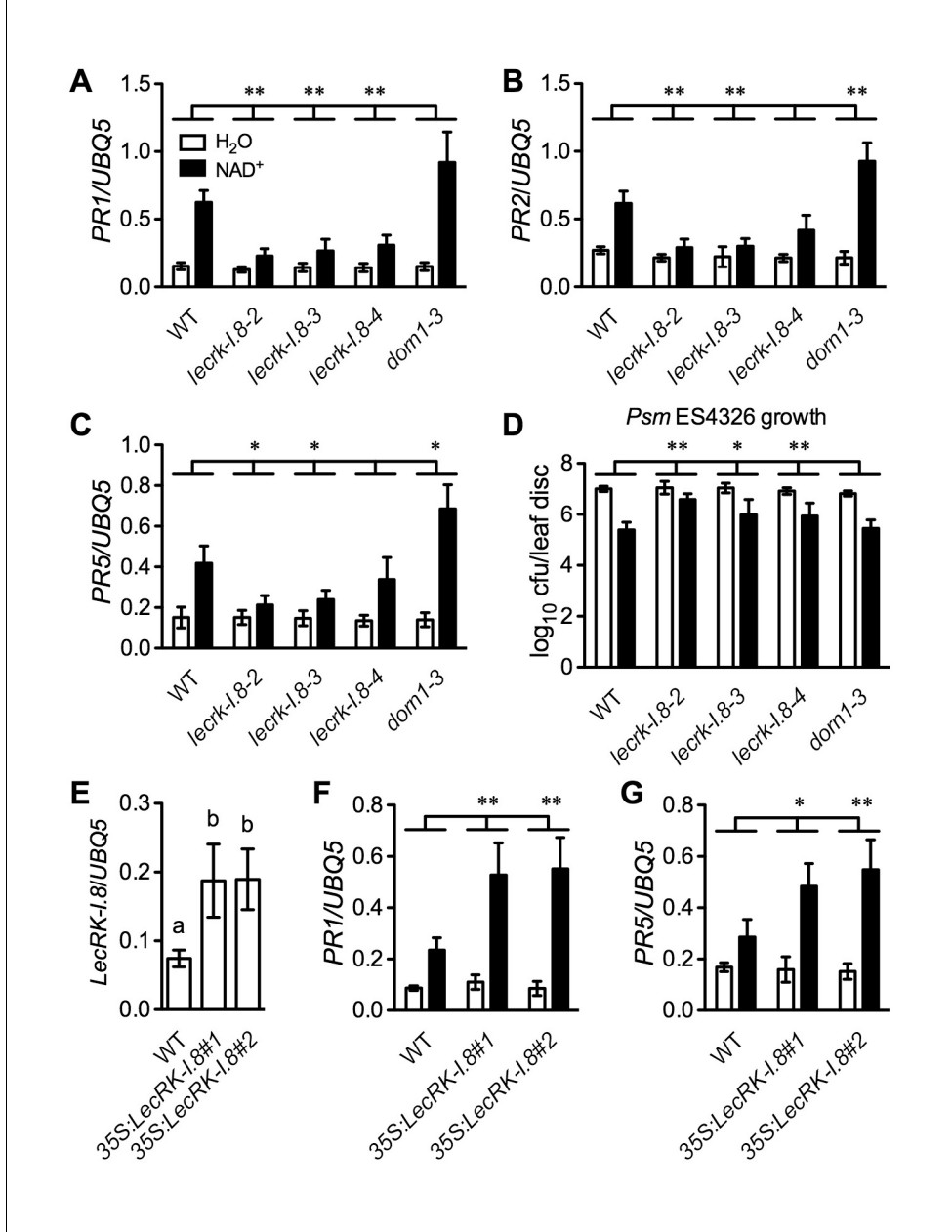

**Figure 2.** LecRK-I.8 functions in extracellular $NAD^+$-triggered defense signaling pathway. (A) to (C) $NAD^+$-induced expression of *PR1* (A), *PR2* (B), and *PR5* (C) was reduced in the *lecrk-I.8* mutants. Plants were treated with 0.2 mM $NAD^+$ solution or water. Leaf tissues were collected 20 hr later for qPCR analysis. Expression levels were normalized against *UBQ5*. Data represent the mean of three independent samples with standard deviation (SD). Asterisks indicate significant differences between the wild type (WT) and the mutants (*p<0.05, **p<0.01, two-way ANOVA). (D) $NAD^+$-induced resistance to the bacterial pathogen *Psm* ES4326 was decreased in the *lecrk-I.8* mutants. Plants were treated as in (A). Five h later, the plants were inoculated with a *Psm* ES4326 suspension ($OD_{600}$ = 0.001). The bacterial titers were determined 3 d post-inoculation. Data represent the mean of eight independent samples with SD. Asterisks indicate significant differences between the wild type and the mutants (*p<0.05, **p<0.01, two-way ANOVA). Cfu: colony-forming units. (E) Expression levels of *LecRK-I.8* were elevated in two *35S:LecRK-I.8* transgenic lines. Data represent the mean of three independent samples with SD. Different letters above the bars indicate significant differences (p<0.05, one-way ANOVA). (F) and (G) Induction of *PR1* (F) and *PR5* (G) by $NAD^+$ was enhanced in the *35S:LecRK-I.8* lines. The experiments were performed as in (A) except that the plants were treated with 0.1 mM $NAD^+$. All experiments were repeated three times with similar trends.

*Figure 2 continued on next page*

*Figure 2 continued*

The following source data and figure supplements are available for figure 2:

**Source data 1.** LecRK-I.8 functions in extracellular NAD$^+$-triggered defense signaling pathway.
**Figure supplement 1.** Exogenous NAD$^+$-induced *Psm* ES4326 resistance in T-DNA insertion lines of 14 candidate genes and transcript levels of *LecRK-I.8* in three T-DNA insertion lines.
**Figure supplement 2.** Characterization of *35S:LecRK-I.8* transgenic lines.

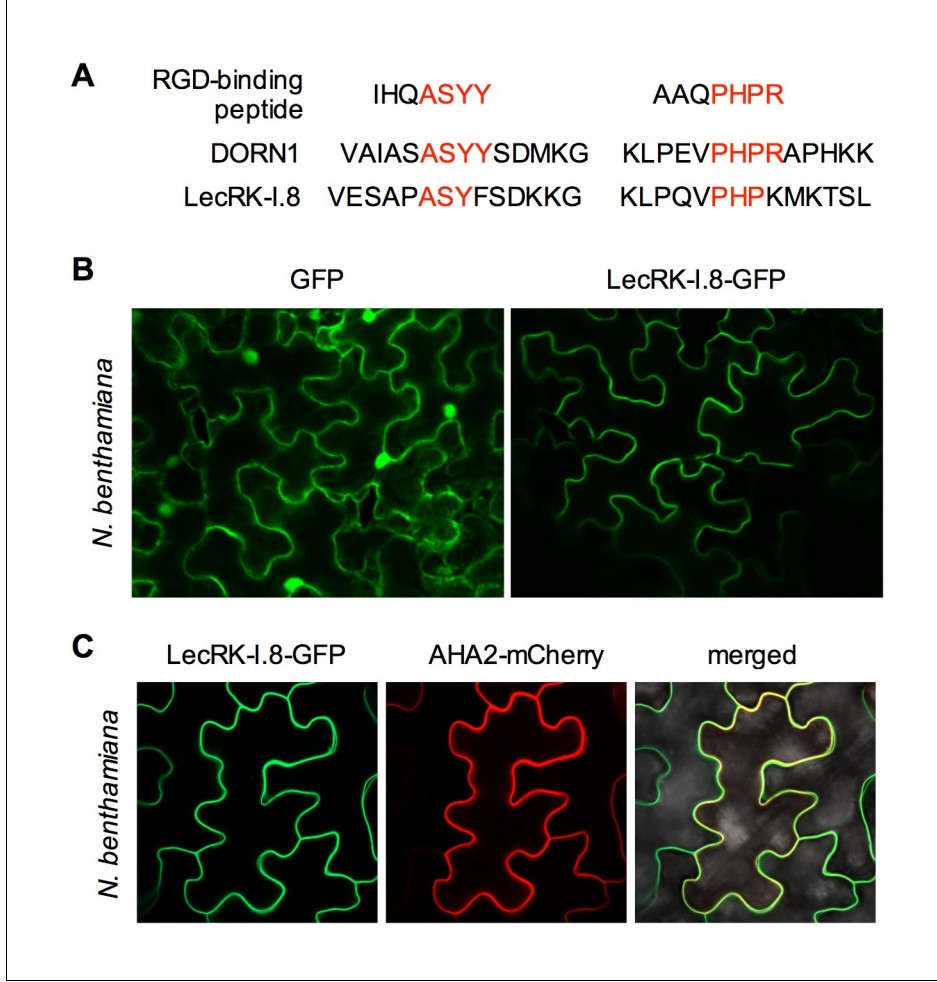

**Figure 3.** Subcellular localization of the LecRK-I.8-GFP fusion protein. (**A**) Putative RGD-binding motifs in DORN1 and LecRK-I.8. (**B**) Confocal images of *N. benthamiana* epidermal cells transiently expressing GFP (Left) and LecRK-I.8-GFP (right). (**C**) Confocal images of *N. benthamiana* epidermal cells transiently co-expressing LecRK-I.8-GFP and AHA2-mCherry. Left: LecRK-I.8-GFP, middle: AHA2-mCherry, and right: merged image.
The following figure supplements are available for figure 3:

**Figure supplement 1.** Alignment between the NAD receptor LecRK-I.8 and the ATP receptor DORN1.
**Figure supplement 2.** Characterization of *35S:LecRK-I.8-GFP lecrk-I.8–2* transgenic lines.

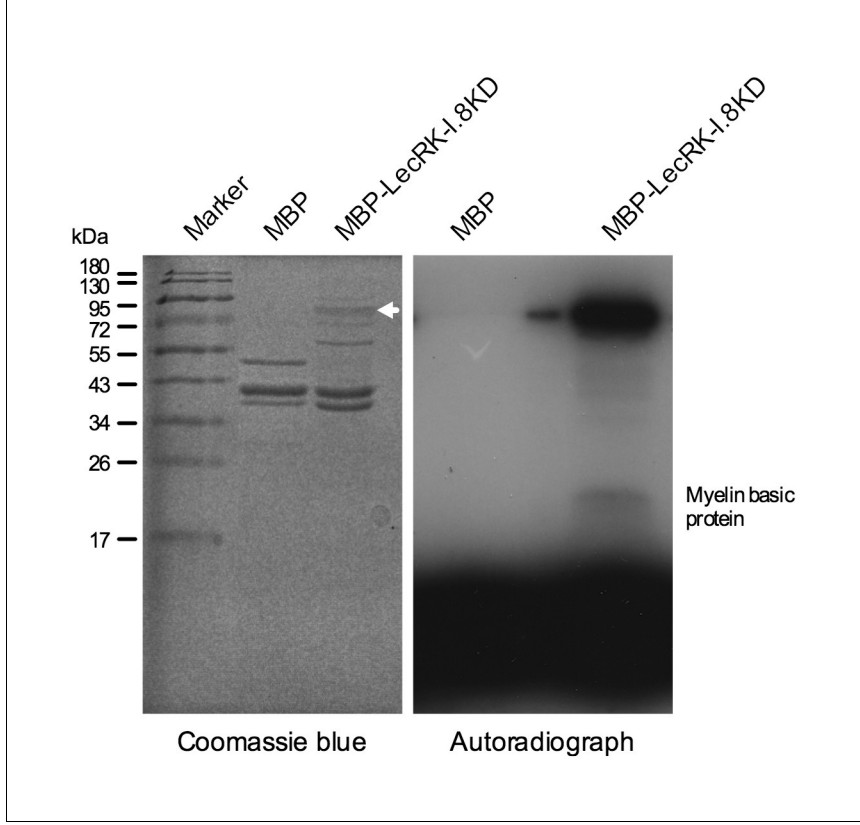

**Figure 4.** Kinase activity of the LecRK-I.8 kinase domain. An autoradiograph (right panel) showing that the kinase domain (KD) of LecRK-I.8 is active based on autophosphorylation and phosphorylation of the myelin basic protein. The purified MBP and MBP-LecRK-I.8KD proteins used for kinase activity assays were separated in a different SDS-PAGE gel (left panel), and the white arrow in the left panel indicates the expected size of the MBP-LecRK-I.8KD protein band. The experiment was repeated with similar results.

immunoprecipitated from transgenic plants expressing GFP using the same antibody (*Figure 5A*), indicating that the NAD$^+$ binding activity is likely specific to eLecRK-I.8. A specific NAD$^+$ binding activity was also detected for an eLecRK-I.8-HA-His fusion protein transiently expressed in *N. benthamiana* (*Figure 5B* and *Figure 5—figure supplement 1B*). Furthermore, NAD$^+$ binding activity was detected for the recombinant fusion protein MBP-eLecRK-I.8 but not for MBP, MBP-eDORN1 (AAs 22 to 288), and MBP-eLecRK-I.3 (AAs 22 to 286) (*Figure 5C* and *Figure 5—figure supplement 1C*). The extracellular domain of LecRK-I.6 (AAs 22 to 286), which is the closest homolog of LecRK-I.8 (*Bouwmeester and Govers, 2009*), did not bind NAD$^+$, either (*Figure 5—figure supplement 2*), suggesting that LecRK-I.6 may not be an eNAD$^+$ receptor. Since LecRK-I.8, DORN1, LecRK-I.3, and LecRK-I.6, all are L-type LecRKs and belong to the LecRK-I clade (*Bouwmeester and Govers, 2009*), this result indicates that not all of the extracellular lectin domains of L-type LecRKs can bind NAD$^+$. In addition, we detected a dramatic increase of NAD$^+$ binding activity in the membrane fractions of the *35S:LecRK-I.8* transgenic plants and a clear decrease of NAD$^+$ binding activity in the membrane fractions of the *lecrk-I.8–2* mutant plants (*Figure 5D*). These results are consistent with the increased and decreased NAD$^+$ responsiveness in the *35S:LecRK-I.8* and *lecrk-I.8–2* plants, respectively (*Figure 2*). Interestingly, there were differences in ligand affinities between the membrane fractions of the *35S:LecRK-I.8* plants and the wild type, which might suggest possible conformation changes when LecRK-I.8 is overexpressed in plants. Finally, the immunoprecipitated eLecRK-I.8-GFP protein showed a typical saturation curve for NAD$^+$ binding with a dissociation constant ($K_d$) of 436.5 ± 104.8 nM (*Figure 5E*), which falls well below the extracellular NAD(H) concentration (~0.4 mM) in pathogen-infected leaf tissues (*Zhang and Mou, 2009*), and thus indicates a relatively high affinity. On the other hand, we found that 5 µM of NAD$^+$ was able to significantly induce the early

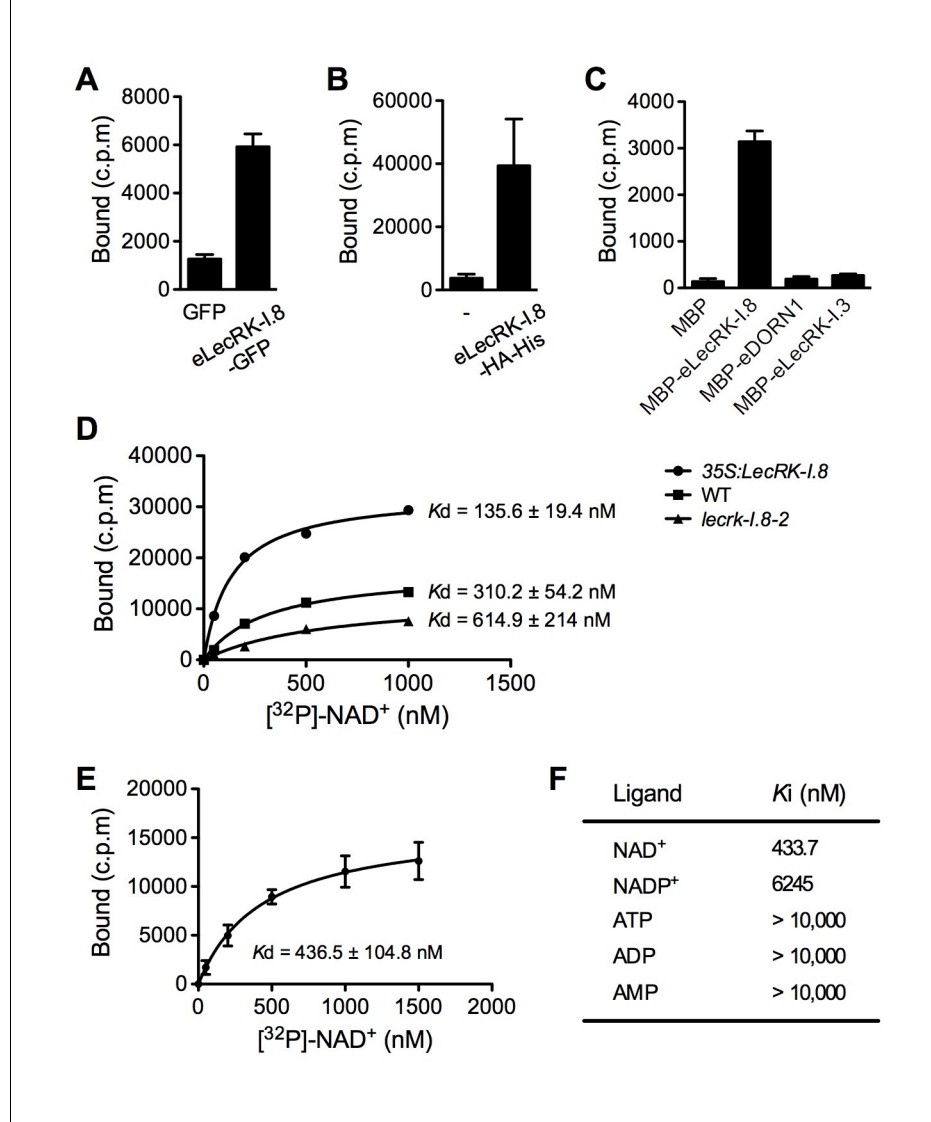

**Figure 5.** LecRK-I.8 binds NAD[+]. (A) to (C), Binding of [32]P-labeled NAD[+] to immunoprecipitated GFP and eLecRK-I.8-GFP proteins (A), purified eLecRK-I.8-HA-His protein (B), and recombinant MBP, MBP-eLecRK-I.8, MBP-eDORN1, and MBP-eLecRK-I.3 proteins (C). (-) in (B) is an empty vector control. Approximately 0.5 g *Arabidopsis* leaf tissues, 1 g *N. benthamiana* leaf tissues, and ~5 μg recombinant proteins were used for each binding assay in (A), (B), and (C), respectively. (D) Binding of [32]P-labeled NAD[+] to the microsomal fractions of *35S:LecRK-I.8*, wild-type (WT), and *lecrk-I.8–2* plants. Specific binding was determined by subtracting the binding in the presence of 1000-fold unlabeled NAD[+] from the total binding in the absence of cold competitor. (E) Saturation binding assay for LecRK-I.8. Immunoprecipitated eLecRK-I.8-GFP proteins were incubated with the indicated concentrations of [32]P-labeled NAD[+] for 30 min. Free NAD[+] was removed by washing. Data were plotted as a specific binding with SD of three experiments. The dissociation constant (*K*d) was calculated by one site specific binding saturation model using GraphPad Prism 5 (www.graphpad.com). (F) Competitive binding assay for LecRK-I.8. Samples containing 250 nM of [32]P-labeled NAD[+] in the presence of 100 nM to 1 mM of unlabeled nucleotides were assayed for specific binding of [32]P-labeled NAD[+]. Inhibition constant (*K*i) values were calculated in GraphPad Prism 5 using the one site Fit *K*i competition model. In (A), (B), (C), and (E), results from three independent experiments were combined (error bars represent SD).

The following source data and figure supplements are available for figure 5:

**Source data 1.** LecRK-I.8 binds NAD[+].

**Figure supplement 1.** Purified eLecRK-I.8 proteins.

*Figure 5 continued on next page*

*Figure 5 continued*

**Figure supplement 2.** NAD$^+$ binding assay of the recombinant MBP-eLecRK-I.6 protein.

**Figure supplement 3.** Induction of several early PAMP-responsive genes by low concentrations of NAD$^+$.

PAMP responsive genes *GLUTATHIONE S-TRANSFERASE1* (*GST1*) and *FLG22-INDUCED RECEPTOR-LIKE KINASE1* (*FRK1*) (*Figure 5—figure supplement 3*), indicating that the NAD$^+$ concentrations required to trigger defense responses are significantly higher than the $K_d$ value of LecRK-I.8, which is rather unusual and difficult to reconcile with a primary ligand sensor function of LecRK-I.8.

However, binding of NAD$^+$ to LecRK-I.8 suggests that LecRK-I.8 may be a receptor for eNAD$^+$. To test the specificity of the binding between NAD$^+$ and LecRK-I.8, we analyzed the ability of various unlabeled nucleotides to compete for binding of $^{32}$P-labeled NAD$^+$. While unlabeled NAD$^+$ exhibited strong competition for binding with $^{32}$P-labeled NAD$^+$, other nucleotides including NADP$^+$, ATP, ADP, and AMP showed little or no competition (*Figure 5F*). Given the structural similarity between NAD$^+$ and NADP$^+$ and the fact that NADP$^+$ can similarly induce immune responses in plants (*Zhang and Mou, 2009*), it is surprising that NADP$^+$ did not efficiently compete for binding with $^{32}$P-labeled NAD$^+$ (*Figure 5F*). These results suggest that LecRK-I.8 may be a receptor specific for NAD$^+$.

## Mutations in *LecRK-I.8* particularly inhibit NAD$^+$-induced defense responses

If LecRK-I.8 is a receptor specific for NAD$^+$, mutations in *LecRK-I.8* should only affect NAD$^+$-induced defense signaling. To test this hypothesis, we treated the *lecrk-I.8* mutants and wild type with NAD$^+$ and three other defense inducers, NADP$^+$, flg22 (a peptide corresponding to the 22 AAs of the conserved N-terminal part of flagellin), and SA. As shown in *Figure 6A and B*, while NAD$^+$ triggered significantly lower levels of *PR1* gene expression and *Psm* ES4326 resistance in the *lecrk-I.8* mutants than those in the wild type, NADP$^+$, flg22, and SA induced similar levels of *PR1* gene expression and *Psm* ES4326 resistance in the *lecrk-I.8* mutants and the wild type. These results indicate a specific role for LecRK-I.8 in eNAD$^+$-triggered signaling. Moreover, LecRK-I.8 has previously been shown to mediate *Pieris brassicae* egg extract-triggered *PR1* gene expression (*Gouhier-Darimont et al., 2013*), but the identity of the elicitor(s) in the egg extracts is unclear. To test if one of the elicitors in insect egg extracts is NAD$^+$, we treated the previously generated *35S:CD38* transgenic plants (*Zhang and Mou, 2012*), *lecrk-I.8–2*, and wild type with *Trichoplusia ni* (cabbage looper) egg extracts following the published protocol (*Gouhier-Darimont et al., 2013*), and analyzed the egg extract-induced *PR1* expression. CD38 is a human NAD(P)-metabolizing ectoenzyme and has been shown to partially block exogenous NAD$^+$-induced *PR1* expression and *Psm* ES4326 resistance (*Zhang and Mou, 2012*). Similarly to *P. brassicae* egg extracts (*Gouhier-Darimont et al., 2013*), *T. ni* egg extracts induced *PR1* gene expression in the wild-type plants, and the induction was dramatically reduced in the *lecrk-I.8–2* mutant (*Figure 6C*). Importantly, *T. ni* egg extract-induced *PR1* expression was also significantly inhibited in the *35S:CD38* plants (*Figure 6C*). Therefore, insect egg extracts may either contain NAD$^+$ as part of their defense inducing activity or induce release of cellular NAD$^+$ into the extracellular space. Taken together, our results strongly suggest that LecRK-I.8 is a potential receptor for NAD$^+$.

## LecRK-I.8 is not the sole mechanism perceiving eNAD$^+$ in *Arabidopsis*

Our results so far have shown that mutations in *LecRK-I.8* partially compromised a low concentration of NAD$^+$-induced *PR* gene expression and *Psm* ES4326 resistance. Since *PR* genes are late defense genes, we compared NAD$^+$-induced expression of several early defense genes, including *AZELAIC ACID INDUCED1* (*AZI1*), *PHYTOALEXIN DEFICIENT4* (*PAD4*), *NON-INDUCIBLE IMMUNITY1* (*NIM1*)-*INTERACTING1* (*NIMIN1*), *NIMIN2*, *WRKY18*, and *WRKY54*, in wild type and the *lecrk-I.8–2* mutant. As shown in *Figure 7—figure supplement 1*, induction of these early defense genes by NAD$^+$ was not significantly affected by the *lecrk-I.8–2* mutation. This result indicates that NAD$^+$

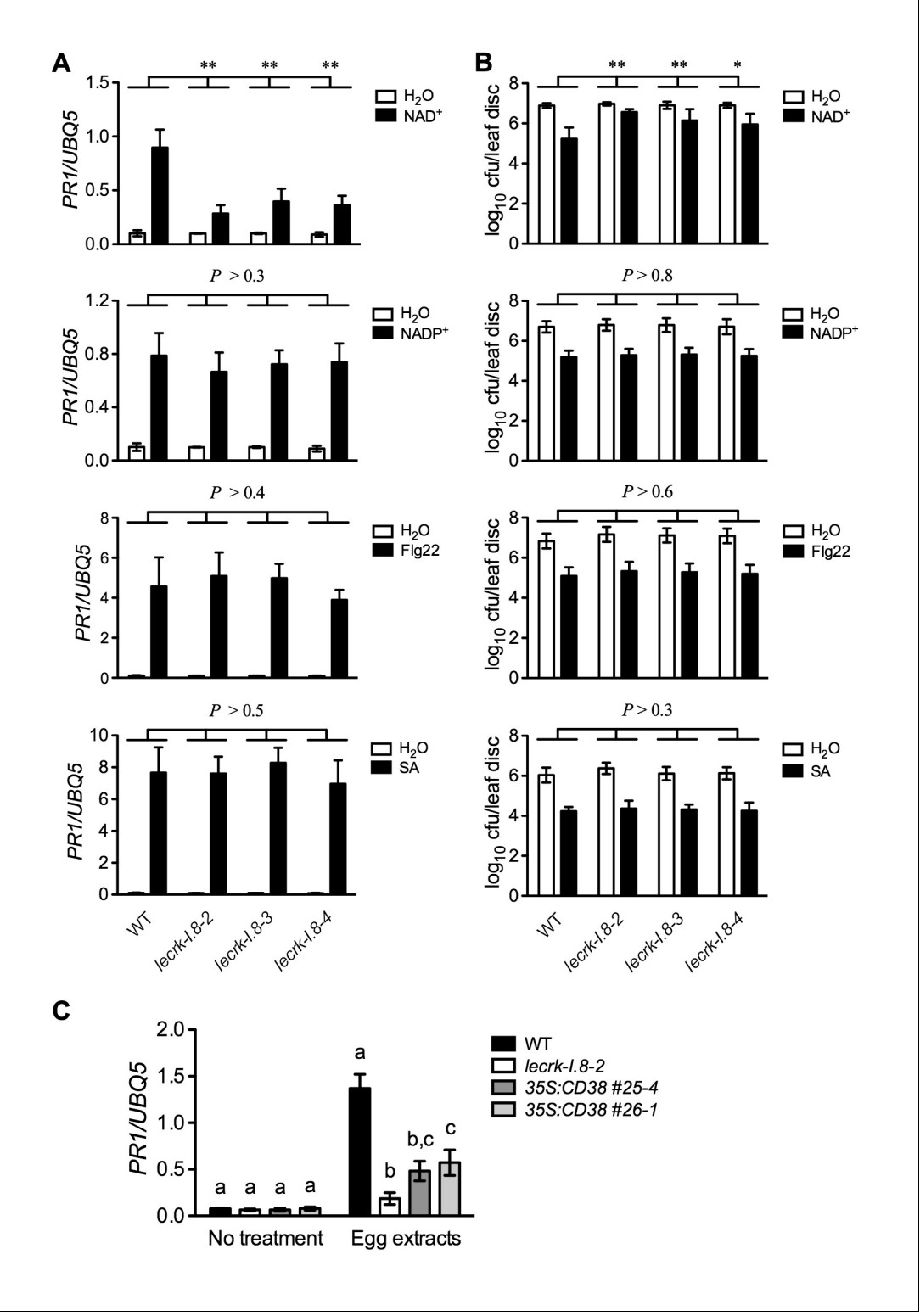

**Figure 6.** Extracellular NADP$^+$-, flg22-, and SA-induced immune responses are not affected in the *lecrk-I.8* mutants. (A) NAD$^+$-, NADP$^+$-, flg22-, and SA-induced *PR1* expression in the wild type (WT) and the *lecrk-I.8* mutants. Plants were infiltrated with 0.2 mM NAD$^+$, 0.2 mM NADP$^+$, 1 µM flg22, or water. For SA treatment, plants were treated with soil drenches plus foliar sprays of 0.5 mM SA solution or water. Leaf tissues were collected 20 hr later for qPCR analysis. Expression levels were normalized against *UBQ5*. Data represent the mean of three independent samples with SD. Asterisks indicate significant differences between the wild type (WT) and the

*Figure 6 continued on next page*

*Figure 6 continued*

mutants (**$p<0.01$, two-way ANOVA). (B) NAD$^+$-, NADP$^+$-, flg22-, and SA-induced *Psm* ES4326 resistance in the wild type and the *lecrk-I.8* mutants. Plants were treated as in (A). Five h after NAD$^+$ or NADP$^+$ treatment and 24 hr after flg22 or SA treatment, the plants were inoculated with a *Psm* ES4326 suspension (OD$_{600}$ = 0.001). The bacterial titers were determined 3 d post-inoculation. Data represent the mean of eight independent samples with SD. Asterisks indicate significant differences between the wild type and the mutants (*$p<0.05$, **$p<0.01$, two-way ANOVA). (C) Insect egg extract-induced *PR1* gene expression in *35S:CD38* transgenic plants. Two μL of *T. ni* egg extracts were dropped onto leaves of the WT, *lecrk-I.8–2*, and *35S:CD38* transgenic plants. The treated leaves without petiole were collected 3 d later for qPCR analysis. Leaf tissues from untreated plants were used as the control. Data represent the mean of three independent samples with SD. Different letters above the bars indicate significant differences ($p<0.05$, one-way ANOVA). The comparison was made separately for each treatment. All experiments were repeated three times with similar trends.

The following source data is available for figure 6:

**Source data 1.** Extracellular NADP$^+$-, flg22-, and SA-induced immune responses are not affected in the *lecrk-I.8* mutants.

perception mechanisms other than LecRK-I.8 still exist in the *lecrk-I.8–2* mutant. To substantiate this conclusion, we treated wild type and the *lecrk-I.8–2* mutant with different concentrations of NAD$^+$ and tested NAD$^+$-induced defense responses. The expression of *PR* genes and resistance to *Psm* ES4326 activated by 0.2 and 0.4 mM NAD$^+$ were significantly reduced in the *lecrk-I.8–2* plants compared with those in the wild type, whereas 0.6 mM NAD$^+$ induced similar levels of *PR* gene expression and *Psm* ES4326 resistance in the mutant and the wild type (*Figure 7*). These results are consistent with the remaining NAD$^+$ binding activity in the membrane fractions of the *lecrk-I.8–2* mutant, supporting the existence of other NAD$^+$ receptors and/or perception mechanisms in *Arabidopsis*.

## LecRK-I.8 contributes to plant basal immunity

If eNAD$^+$ is an endogenous signaling molecule and LecRK-I.8 is an essential cell-surface receptor for eNAD$^+$, mutations in *LecRK-I.8* should compromise immune responses. However, in *Figures 2D*, *6B* and *7B*, where the plants were infected with *Psm* ES4326 at a high dose (an inoculum of OD$_{600}$ = 0.001) (*Clarke et al., 2000*), the bacterial pathogen grew to similar levels in mock (water)-treated wild type and *lecrk-I.8* mutants. The high dose is generally used for disease resistance test and may not be able to resolve differences in disease susceptibility (*Glazebrook et al., 1996*; *Clarke et al., 2000*). Therefore, we inoculated wild-type and *lecrk-I.8* plants with a low dose (an inoculum of OD$_{600}$ = 0.0001). Under this condition, both *lecrk-I.8–2* and *lecrk-I.8–3* exhibited significantly reduced *PR1* gene induction and significantly enhanced susceptibility to *Psm* ES4326, compared with the wild type and the *dorn1-3* mutant (*Figure 8*). We also tested biological induction of SAR in the *lecrk-I.8* mutants and found that SAR induction was not affected in all three *lecrk-I.8* alleles (*Figure 8—figure supplement 1*). Nevertheless, our results demonstrate that the potential eNAD$^+$ receptor LecRK-I.8 plays a positive role in plant immunity.

## Discussion

Here, we present several lines of evidence to demonstrate that LecRK-I.8 is a potential eNAD$^+$ receptor in *Arabidopsis*. First, the *LecRK-I.8* gene is induced by exogenous NAD$^+$ (*Supplementary file 1B*), which is consistent with the observation that receptor-encoding genes are often ligand inducible (*Zipfel et al., 2006*). Second, LecRK-I.8 is localized in the plasma membrane and possesses an active cytoplasmic kinase domain (*Figures 3* and *4*). Third, LecRK-I.8 binds NAD$^+$, but not NADP$^+$, ATP, ADP, or AMP (*Figure 5F*), and three other LecRKs, DORN1, LecRK-I.3, and LecRK-I.6, do not bind NAD$^+$ (*Figure 5C* and *Figure 5—figure supplement 2*). Fourth, mutations in *LecRK-I.8* inhibit NAD$^+$-induced, but not NADP$^+$-, flg22-, and SA-induced, defense responses (*Figure 6A and B*). Finally, mutations in *LecRK-I.8* compromise basal resistance to the bacterial pathogen *Psm* ES4326 (*Figure 8*).

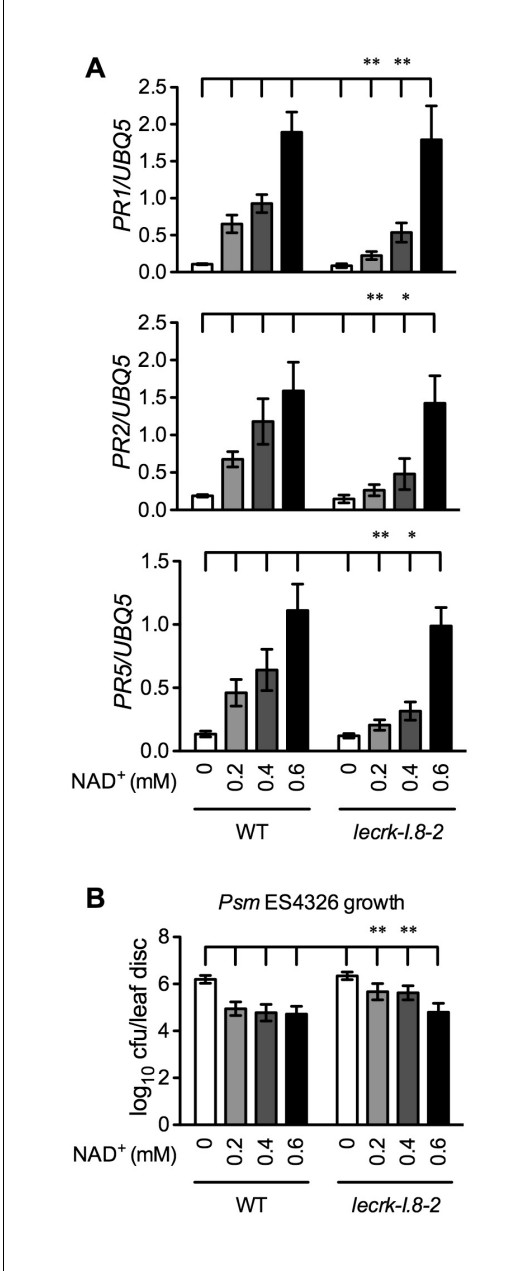

**Figure 7.** Immune responses induced by different concentrations of NAD$^+$ in *lecrk-I.8–2*. (**A**) Comparison of different concentrations of NAD$^+$-induced expression of *PR1*, *PR2*, and *PR5* in *lecrk-I.8–2* and the wild type (WT). Leaves of 4-week-old soil-grown plants were infiltrated with the indicated concentrations of NAD$^+$. Total RNA was extracted from the infiltrated leaves 20 hr later and subjected to real-time qPCR analysis. Expression was normalized against constitutively expressed *UBQ5*. Data represent the mean of three independent samples with SD. Asterisks indicate significant differences between *lecrk-I.8–2* and the wild type (*$p < 0.05$, **$p < 0.01$, two-way ANOVA). (**B**) Comparison of different concentrations of NAD$^+$-induced resistance to *Psm* ES4326 in *lecrk-I.8–2* and the

*Figure 7 continued on next page*

NAD$^+$ has long been shown to bind to rat brain synaptic membranes (*Khalmuradov et al., 1983*), and recent studies have also indicated that several purinergic P2X and P2Y receptors are involved in eNAD$^+$-induced biological responses (*Moreschi et al., 2006*; *Mutafova-Yambolieva et al., 2007*; *Grahnert et al., 2009*; *Klein et al., 2009*). However, there has been no direct evidence demonstrating the binding of NAD$^+$ to a known cell-surface receptor. Therefore, the identity of eNAD$^+$-binding receptors has remained unclear. Our identification of a LecRK as a potential eNAD$^+$ receptor suggests that lectin receptors may directly bind NAD$^+$, leading to transmembrane signaling. As lectin receptors are widely distributed in both the animal and plant kingdoms (*Drickamer and Taylor, 1993*; *Vaid et al., 2012*), and certain animal and plant lectin domains have convergently evolved similar ligand binding architecture (*Loris, 2002*), animal cells might also use lectin receptors to sense eNAD$^+$. This speculation could be tested using receptor-NAD$^+$ binding assays.

Identification of a potential eNAD$^+$ receptor in *Arabidopsis* provides direct evidence for eNAD$^+$ being a bona fide endogenous signaling molecule in plants (*Zhang and Mou, 2012*). This is consistent with the fact that genes involved in plant immune responses are significantly enriched in the genes upregulated by NAD$^+$ and that the majority of the NAD$^+$-induced genes are also activated by the bacterial pathogen *Pst* DC3000/*avrRpt2* (*Figure 1*) (*Wang et al., 2013*). These results together should eradicate the skepticism concerning the physiological relevance of eNAD$^+$ perception in plants (*Fu and Dong, 2013*). Additionally, given the diverse roles played by eNAD$^+$ in animal cells (*Billington et al., 2006*; *Iqbal and Zaidi, 2006*; *Haag et al., 2007*; *Adriouch et al., 2012*; *Mutafova-Yambolieva and Durnin, 2014*), further studies with the *Arabidopsis lecrk-I.8* mutants will likely reveal new biological functions for this signaling molecule in plants.

eNAD$^+$ may play a role in plant-insect interactions. It has recently been reported that deposition of *P. brassicae* egg batches on *Arabidopsis* leaves induces the SA signaling pathway, which in turn suppresses the JA pathway, thus benefiting the insect progeny (*Little et al., 2007*; *Bruessow et al., 2010*). By treatment with *P. brassicae* egg extracts, which mimics the effect of oviposition, *Gouhier-Darimont et al. (2013)* identified LecRK-I.8 as a potential cell surface receptor for the insect egg-derived elicitors. Although it has been shown that a fraction from purified *P. brassicae* egg lipids is able to induce

*Figure 7 continued*
wild type. Leaves of 4-week-old soil-grown plants were infiltrated with the indicated concentrations of $NAD^+$. Five h later, the infiltrated leaves were inoculated with a *Psm* ES4326 suspension ($OD_{600}$ = 0.001). The in planta bacterial titers were determined 3 d post-inoculation. Data represent the mean of eight independent samples with SD. Asterisks indicate significant differences between *lecrk-I.8–2* and the wild type (\*\*p<0.01, two-way ANOVA). Experiments were repeated three times with similar trends.
The following source data and figure supplement are available for figure 7:

**Source data 1.** Immune responses induced by different concentrations of $NAD^+$ in *lecrk-I.8-2*.
**Figure supplement 1.** $NAD^+$-induced expression of several early defense-responsive genes in the *lecrk-I.8–2* mutant.

*PR1* gene expression (*Gouhier-Darimont et al., 2013*), the identity of the elicitors still remains unknown. Three pieces of evidence generated in this study indicate that the defense-inducing activity of the insect egg extract could be, at least, partially attributed to $NAD^+$, and/or that the perception of egg extract may lead to release of cellular $NAD^+$ into the extracellular space. First, LecRK-I.8 is potentially a cell surface receptor specific for $NAD^+$ (*Figure 5*). Second, the human NAD(P)-metabolizing ectoenzyme CD38 inhibits *T. ni* egg extract-induced *PR1* gene expression (*Figure 6C*). Finally, the majority of *P. brassicae* oviposition-induced receptor-like kinase genes (*Little et al., 2007*) are also upregulated by exogenous $NAD^+$ addition (*Supplementary file 1C*). Thus, insect eggs appear to use $NAD^+$ to alter the SA-JA signaling balance in plants for the benefit of the insect progeny.

It has been shown that $eNAD^+$ and eATP play multiple, partially overlapping roles in animal immune cells (*Haag et al., 2007*). One of the best-studied examples is the mechanism activating the $P2X_7$ purinoceptor by $eNAD^+$ and eATP in T cells and macrophages (*Bartlett et al., 2014*; *Rissiek et al., 2015*). While ATP activates $P2X_7$ through direct binding (*Surprenant et al., 1996*; *Rassendren et al., 1997*; *Chessell et al., 1998*), $NAD^+$ regulates $P2X_7$ in these two different cell types via distinct mechanisms. In T cells, $NAD^+$ promotes ART-mediated ADP-ribosylation of $P2X_7$, which is sufficient for activation of the purinoceptor (*Seman et al., 2003*; *Adriouch et al., 2008*), whereas in macrophages, ADP-ribosylation does not activate $P2X_7$ but rather reduces the threshold concentration of ATP needed to turn on the receptor (*Hong et al., 2009*). Thus, $eNAD^+$ and eATP function additively and synergistically in T cells and macrophages, respectively, to regulate $P2X_7$ signaling. In contrast, available evidence obtained by studies in *Arabidopsis* indicates that $eNAD^+$ and eATP act antagonistically to modulate the SA signaling pathway. $eNAD^+$ induces SA accumulation and SA-dependent defense gene expression and disease resistance (*Zhang and Mou, 2009*), whereas eATP suppresses these defense responses (*Chivasa et al., 2009*). In agreement with these results, induction of *PR* genes by exogenous $NAD^+$ in the eATP receptor mutant *dorn1-3* is significantly enhanced compared with that in the wild type (*Figure 2A–C*). On the other hand, the *dorn1-3* mutation does not significantly affect basal and $NAD^+$-induced resistance to the bacterial pathogen *Psm* ES4326 (*Figures 2D* and *8B*), which may be attributed to the presence of other eATP receptors. Whether knockout of *LecRK-I.8* influences eATP-mediated suppression of SA signaling needs further investigation.

As in animal cells (*Ziegler and Niere, 2004*), $eNAD^+$ is likely perceived by multiple receptors and/or mechanisms in plants. Mutations in *LecRK-I.8* only block 0.2 and 0.4 mM $NAD^+$-induced, but not 0.6 mM $NAD^+$-induced, *PR* gene expression and disease resistance (*Figure 7*). Furthermore, induction of several early defense responsive genes by 0.2 mM $NAD^+$ is not inhibited by the *lecrk-I.8–2* mutation (*Figure 7—figure supplement 1*). These results indicate that LecRK-I.8 is not the sole $eNAD^+$ perception mechanism in *Arabidopsis*. Indeed, the *Arabidopsis* genome contains genes encoding 75 LecRKs (32 G-type, 42 L-type, and 1 C-type) as well as a large number of other RKs and channels (*Arabidopsis Genome Initiative, 2000*; *Vaid et al., 2012*), many of which are also $NAD^+$ inducible and could potentially encode $eNAD^+$ receptors. Thus, a genome-wide survey via $NAD^+$ binding assays is warranted for identification of other $eNAD^+$ receptors in *Arabidopsis*.

It is currently unclear whether $eNAD^+$ signaling is specific for *Arabidopsis* or the mustard family (Brassicaceae). Recent genome-wide analyses of LecRKs in multiple plant species revealed that this large family of RKs also exists in other dicots and monocots. For instance, there are 231 LecRKs (180 G-type, 50 L-type, and 1 C-type) in *Populus* (*Yang et al., 2016*), 173 LecRKs (100 G-type, 72 L-type, and 1 C-type) in rice (*Vaid et al., 2012*), 263 LecRKs (177 G-type, 84 L-type, and 2 C-type) in bread

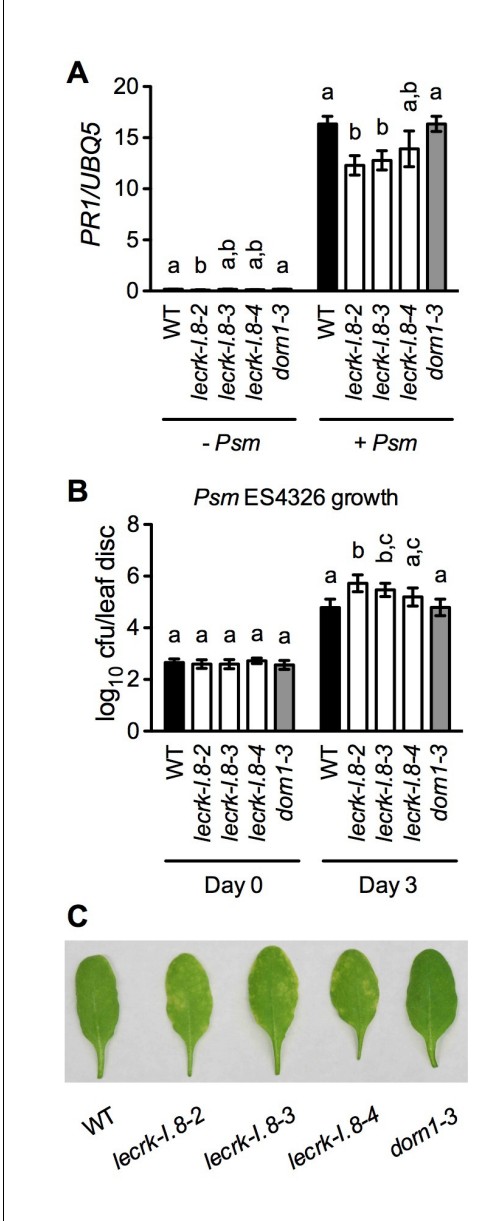

**Figure 8.** Basal immunity is compromised in the *lecrk-I.8* mutants. (**A**) *Psm* ES4326-induced *PR1* expression was inhibited in the *lecrk-I.8* mutants. Plants were inoculated with (+) or without (−) a *Psm* ES4326 suspension ($OD_{600} = 0.0001$). Leaf tissues were collected 24 hr post-inoculation for qPCR analysis. Expression was normalized against constitutively expressed *UBQ5*. Data represent the mean of three independent samples with SD. Different letters above the bars indicate significant differences ($p<0.05$, one-way ANOVA). The comparison was made separately for each treatment. WT: wild type. (**B**) and (**C**), The *lecrk-I.8* mutants were more susceptible to *Psm* ES4326 than the wild type. Plants were inoculated with a *Psm* ES4326 suspension ($OD_{600} = 0.0001$). The bacterial titers in (**B**) were determined immediately and 3 d post-
*Figure 8 continued on next page*

wheat (*Shumayla et al., 2016*), and 113 LecRKs (59 G-type, 53 L-type, and 1 C-type) in foxtail millet (*Zhao et al., 2016*). These results strongly suggest that $eNAD^+$ receptors may be broadly distributed in the plant kingdom. Further investigations are needed to test if $eNAD^+$ signaling is in play in diverse plant species and to identify $eNAD^+$ receptors in these plant species.

## Materials and methods

### Plant materials and growth conditions

The wild type used in this study was the *Arabidopsis thaliana* (L.) Heynh. ecotype Columbia (Col-0). The T-DNA insertion lines used in this study are listed in *Supplementary file 1B*, and the *dorn1-3* mutant (SALK_042209) was previously described (*Choi et al., 2014*). The T-DNA insertion lines were obtained from either Arabidopsis Biological Resource Center at The Ohio State University (Columbus, OH) or Nottingham Arabidopsis Stock Center at The University of Nottingham (Nottingham, UK). Homozygous mutant plants of the T-DNA insertion lines were confirmed with primers flanking the T-DNA insertions and the left border primers LBa1, LB3, and o8409 (*Supplementary file 1D*). The Arabidopsis seeds were sown on autoclaved soil (Sunshine MVP; Sun Gro Horticulture, Agawam, MA) and cold-treated at 4°C for 3 days. Plants were germinated and grown at 22°C to 24°C under a 16-hr-light/8-hr-dark regime. Four-week-old soil-grown plants were used for chemical treatment and pathogen infection.

### Chemical and cabbage looper egg extract treatment

$NAD^+$-Na and $NADP^+$-Na were dissolved in water, and the pH of the resulting solutions was adjusted to ~6.0 using 0.1 M NaOH. Flg22 was dissolved in water to make a 100 µM stock solution, which was freshly diluted before each experiment. For $NAD^+$, $NADP^+$, and flg22 treatment, the solution was infiltrated into *Arabidopsis* leaves using a 1 mL needleless syringe. For SA treatment, plants were soil-drenched with 0.5 mM SA, sprayed with 0.5 mM SA plus 0.1% Tween 20, and partially covered with a transparent plastic dome. Cabbage looper eggs were ordered from Benzon Research Inc. (Carlisle, PA). Generation of egg extracts and treatment with egg extracts were conducted as previously described (*Gouhier-Darimont et al., 2013*).

*Figure 8 continued*

inoculation. Data represent the mean of eight independent samples with SD. Different letters above the bars indicate significant differences (p<0.05, one-way ANOVA). The comparison was made separately for each time point. Photos showing the disease symptoms in (C) were taken 3 d post-inoculation. Experiments in (A) and (B) were repeated three times with similar trends.

The following source data and figure supplement are available for figure 8:

**Source data 1.** Basal immunity is compromised in the *lecrk-I.8* mutants.

**Figure supplement 1.** Biological induction of SAR in *lecrk-I.8* mutants.

## Pathogen infection

Inoculation of *Arabidopsis* plants with the bacterial pathogen *Psm* ES4326 was performed by pressure-infiltration using a 1 mL needleless syringe. Eight leaves per genotype/treatment from eight plants were collected immediately after the inoculation and/or 3 d post-inoculation to examine the growth of the pathogen.

## RNA and protein analysis

Total RNA extraction, reverse transcription, and real-time qPCR were performed as previously described (*Wang et al., 2015*) using gene-specific primers (*Supplementary file 1D*). Leaf tissues for each independent RNA sample were collected from 12 plants. Protein gel blot analysis was conducted as described previously (*Mou et al., 2003*).

## Plasmid construction and plant transformation

For subcellular localization study, the DNA fragments encoding mGFP in pRTL2-mGFP and mCherry in pNDH-OCT were amplified using the primers XbaI-SalI-GFPF/XhoI-GFPR and XbaI-SalI-mCherryF/XhoI-mCherryR, respectively. The primers used for plasmid construction are listed in *Supplementary file 1D*. The PCR products were digested with XbaI and XhoI and cloned into XbaI/SalI-digested pCAMBIA1300S to create pCAMBIA1300S-GFP and pCAMBIA1300S-mCherry. Then the coding regions of *LecRK-I.8* and *AHA2* were amplified using the primers BamHI-flLecRK-I.8F/SalI-flLecRK-I.8R and SacI-AHA2F/SalI-AHA2R, respectively. The PCR products were digested with BamHI or SacI and SalI and cloned into the corresponding sites of pCAMBIA1300S-GFP and pCAMBIA1300S-mCherry, resulting in pCAMBIA1300S-LecRK-I.8-GFP and pCAMBIA1300S-AHA2-mCherry, respectively. For generation of transgenic *Arabidopsis* expressing eLecRK-I.8-GFP, the *LecRK-I.8* fragment encoding the extracellular domain was amplified using the primers EcoRI-ATGLecRK-I.8F and BspHI-eLecRK-I.8R. The PCR products were digested with EcoRI and BspHI and cloned into EcoRI/NcoI-digested pRTL2-mGFP to generate pRTL2-eLecRK-I.8-GFP. Then the 35S:eLecRK-I.8-GFP cassette was recovered using HindIII and subcloned into HindIII-digested and calf intestinal phosphatase-treated pCB302 to produce pCB302-35S:eLecRK-I.8-GFP. For transient expression in *N. benthamiana*, the *LecRK-I.8* fragment encoding the extracellular domain was amplified using the primers SacI-eLecRK-I.8F and SalI-HisHAeLecRK-I.8R. The PCR products were digested with SacI and SalI and cloned into the corresponding sites of the vector pCAMBIA1300S, resulting in the plasmid pCAMBIA1300S-eLecRK-I.8-HA-His. For expression of the MBP-eLecRK-I.8, MBP-eDORN1, MBP-eLecRK-I.3, MBP-eLecRK-I.6, and MBP-LecRK-I.8KD fusion proteins in *Escherichia coli*, DNA fragments encoding the corresponding domains of these proteins were amplified using the primers listed in *Supplementary file 1D*. The PCR products were digested with appropriate restriction enzymes and cloned into the corresponding sites of pMAL-p2X, generating pMAL-p2X-fragment plasmids. The plasmids pCB302-35S:eLecRK-I.8-GFP, pCAMBIA1300S-eLecRK-I.8-HA-His, pCAMBIA1300S-LecRK-I.8-GFP and pCAMBIA1300S-AHA2-mCherry were introduced into the Agrobacterium strain GV3101(pMP90) and the pMAL-p2X-fragment plasmids were introduced into the *E. coli* strain BL21(DE3) by electroporation.

*Arabidopsis* wild-type Col-0 plants were transformed with Agrobacteria carrying the pCB302-35S:eLecRK-I.8-GFP plasmid following the floral dip protocol (*Clough and Bent, 1998*). Transient expression in *N. benthamiana* was performed as described previously with slight modifications (*Krasileva et al., 2010*). Briefly, Agrobacteria carrying pCAMBIA1300S-LecRK-I.8-GFP, pCAMBIA1300S-AHA2-mCherry, or pCAMBIA1300S-eLecRK-I.8-HA-His were suspended in an induction buffer (10 mM MES-KOH, pH 5.6, 10 mM MgCl2 and 100 μM acetosyringone) to an $OD_{600}$ of 0.4, preinduced for 2 to 3 hr at 28°C, and then infiltrated into *N. benthamiana* leaves using a 1 mL

needleless syringe. Two to 3 d later, the leaves infiltrated with the Agrobacteria carrying pCAM-BIA1300S-LecRK-I.8-GFP/pCAMBIA1300S-AHA2-mCherry and pCAMBIA1300S-eLecRK-I.8-HA-His were used for microscopy analysis and protein purification, respectively.

## Protein purification

For immunoprecipation, leaf tissues from 3-week-old soil-grown *35S:eLecRK-I.8-GFP* and *35S:GFP* plants were homogenized on ice in extraction buffer (50 mM HEPES, pH 7.5, 50 mM NaCl, 10 mM EDTA, 5 mM DTT, 1% Triton X-100, and protease inhibitors: 50 µg/mL TPCK, 50 µg/mL TLCK, and 0.6 mM PMSF). The homogenates were centrifuged at 20,000 *g* at 4°C for 20 min and the supernatants were transferred to new Eppendorf tubes. Monoclonal anti-GFP antibodies were added to the extracts (1:200). After incubation at 4°C for 1 hr, the antibody bound proteins were precipitated by adding protein G plus-agarose beads to the extracts (20 µL/mL), followed by incubation at 4°C overnight. The beads were then precipitated by centrifugation at 2000 rpm for 5 min, washed 3 times with the extraction buffer without detergent, and then used for immunoblotting and $NAD^+$ binding assays. For purification of eLecRK-I.8-HA-His, agroinfiltrated *N. benthamiana* leaf tissues were homogenized on ice in extraction buffer (50 mM Tris-HCl, pH7.5, 150 mM NaCl, 0.1% Triton X-100, 0.2% Nonidet P-40, 6 mM $\beta$-mercaptoethanol, and protease inhibitors: 50 µg/mL TPCK, 50 µg/mL TLCK, and 0.6 mM PMSF). The homogenates were centrifuged at 20,000 g at 4°C for 20 min and the supernatants were transferred to new Eppendorf tubes. The eLecRK-I.8-HA-His protein was purified using HisPur Cobalt Resin following the manufacturer's protocol (Thermo Scientific, Waltham, MA). After washing, the resins with bound proteins were used for immunoblotting and $NAD^+$ binding assays. For purification of MBP-eLecRK-I.8, MBP-eDORN1, MBP-eLecRK-I.3, MBP-eLecRK-I.6, and MBP-LecRK-I.8KD, a single colony of BL21(DE3) carrying the corresponding plasmid was cultured overnight at 37°C in 5 mL Lysogeny broth (LB) with 50 µg/mL ampicillin. One mL of the seed culture was added to 500 mL fresh LB medium with 50 µg/mL ampicillin and cultured at 37°C with shaking to an $OD_{600}$ of 0.4. Isopropyl $\beta$-D-1-thiogalactopyranoside was added to a final concentration of 0.3 mM and the culture was allowed to grow for another 16–18 hr at 18°C before the cells were harvested for protein extraction. MBP-fusion proteins were purified with amylose resin according to the protocol supplied by the manufacturer (New England Biolabs, Ipswich, MA).

## $NAD^+$ binding assays

$NAD^+$ binding experiments were performed following a previously described protocol for brassinolide binding assays (*Wang et al., 2001*). Briefly, beads with the bound proteins were re-suspended in binding buffer (10 mM HEPES, pH 7.5, 5 mM $MgCl_2$) and aliquoted in 86 µL portions for individual binding reactions. For total binding assay, 10 µL binding buffer and 4 µL $^{32}P$-labeled $NAD^+$ (6.25 µM) were added, resulting in 250 nM $^{32}P$-labeled $NAD^+$ in the final 100 µL reaction mixture. $^{32}P$-labeled $NAD^+$ (specific activity 800 Ci/mmol) was purchased from PerkinElmer (Waltham, MA). For nonspecific binding, 10 µL of 2.5 mM unlabeled $NAD^+$ and 4 µL $^{32}P$-labeled $NAD^+$ were added, resulting in 250 µM unlabeled $NAD^+$ and 250 nM $^{32}P$-labeled $NAD^+$ in the final 100 µL reaction mixture. The mixtures were incubated for 30 min at room temperature with gentle mixing every 5 min. The beads containing the binding reactions were then precipitated by centrifugation at 2000 rpm for 5 min, washed 3 times with the binding buffer, re-suspended in 10 mL scintillation counter liquid per sample, and carefully transferred into scintillation vials. The vials were placed in a Beckman Coulter LS6500 Multi-Purpose Scintillation Counter (Beckman Coulter, Brea, CA) and bound $^{32}P$-labeled $NAD^+$ was quantified by scintillation counting. Specific $NAD^+$ binding was calculated by subtracting the nonspecific binding from the total binding. For saturation binding assay, protein G plus-agarose beads with the bound proteins were incubated with different concentrations (50, 200, 500, 1000, and 1500 nM) of $^{32}P$-labeled $NAD^+$ in the absence (for total binding) or presence (for nonspecific binding) of additional 1000-fold unlabeled $NAD^+$ in the binding buffer. For competitive binding assay, protein G plus-agarose beads with the bound proteins were incubated with 250 nM of $^{32}P$-labeled $NAD^+$ in the presence of different concentrations (100 nM, 1 µM, 10 µM, 100 µM and 1 mM) of unlabeled nucleotides ($NAD^+$, $NADP^+$, ATP, ADP and AMP) in the binding buffer. The dissociation constant (*K*d) was calculated by one site specific binding saturation model using GraphPad Prism 5 (www.graphpad.com). Inhibition constant ($K_i$) values were calculated by importing the data points into GraphPad Prism 5 (www.graphpad.com) using the one site Fit $K_i$ competition model.

For microsome-based binding assays, microsomes were prepared as previously described with minor modifications (*Wang et al., 2001*; *Caño-Delgado and Wang, 2009*). All steps were conducted at 4°C. Arabidopsis plants were grown under a 15-hr-light/9-hr-dark regime for about 6 weeks. Rosette leaf tissues were homogenized with a mortar and pestle in 1 mL/1 gram chilled membrane extraction buffer (20 mM Tris-HCl, pH 7.5, 250 mM mannitol, 5 mM $MgCl_2$, 0.1 mM $CaCl_2$, and protease inhibitors). Homogenates were filtered through two layers of Miracloth and centrifuged at 10,000 $g$ for 10 min at 4°C. The supernatant was centrifuged at 100,000 $g$ for 1 hr at 4°C to pellet microsomes. The microsomal pellet was resuspended at a protein concentration of 2 mg/mL in binding buffer (10 mM MES-KOH, pH 5.7, 5 mM $MgCl_2$, 0.25 mM $CaCl_2$, 0.25 M mannitol, and protease inhibitors). Each binding assay contains 50 µL microsomes, indicated amount (50, 200, 500, 1000 nM) of $^{32}P$-labeled $NAD^+$, 1 mg/mL BSA, with 1000-fold excess unlabeled $NAD^+$ for background binding assays. The final reaction volume was brought to 100 µL by adding binding buffer. The binding reactions were incubated for 30 min at room temperature with gentle mixing every 5 min. The bound and free $^{32}P$-labeled $NAD^+$ were separated by filtering the mixture through a glass-fibre filter (Whatman) and washing with 10 mL ice-cold binding buffer, and were quantified by scintillation counting. Binding data were analyzed and plotted using GraphPad Prism 5.

## Kinase assay

The kinase assay was performed as described by *Choi et al. (2014)* with minor modifications. Briefly, 5 µg of MBP-LecRK-I.8KD and 2 µg of myelin basic protein in a buffer (50 mM Tris-HCl pH7.5, 50 mM KCl, 10 mM $MnCl_2$, 10 mM $MgCl_2$) plus 1 µL of ATP (0.4 µL $^{32}P$-labeled ATP, 0.4 µL cold ATP, 0.2 µL $H_2O$) in a total volume of 30 µL were incubated at 30°C for 30 min. $^{32}P$-labeled ATP (specific activity 3000 Ci/mmol) was purchased from PerkinElmer. After SDS-PAGE electrophoresis, the gel was dried and exposed to X-ray film for 3 hr.

## Microarray analysis

Four-week-old soil-grown plants were treated with 1 mM $NAD^+$ or water. Total RNA samples extracted from the treated leaves were subjected to microarray analysis. Briefly, RNA concentration was determined on a NanoDrop Spectrophotometer (Thermofisher Scientific, Waltham, MA) and sample quality was assessed using the 2100 Bioanalyzer (Agilent Technologies, Santa Clara, CA). cDNA was synthesized from 200 ng of total RNA and used as a template for in vitro transcription in the presence of T7 RNA Polymerase and cyanine labeled CTP's using the Quick Amp Labeling kit (Agilent Technologies) according the manufacturer's protocol. The amplified, labeled complementary RNA (cRNA) was purified using the RNeasy Mini kit (Qiagen, Valencia, CA). For each array, 1650 ng of Cy three labeled cRNA was fragmented and hybridized with rotation at 65°C for 17 hr. Samples were hybridized to Arabidopsis 4 × 44 k arrays (Agilent Technologies). The arrays were washed according to the manufacturer's protocol and then scanned on a G2505B scanner (Agilent Technologies). Data were extracted using Feature Extraction 10.1.1.1 software (Agilent Technologies).

Data (individual signal intensity values) obtained from the microarray probes were background corrected using a *normexp+offset* method, in which a small positive offset ($k$ = 50) was added to move the corrected intensities away from zero (*Ritchie et al., 2007*). The resulting data were log transformed (using two as the base) and normalized between individual samples using quantile normalization (*Smyth, 2005*). After normalization, a linear model was fitted on each gene for comparison using the *limma* package in R (*Ritchie et al., 2015*). To control false discovery rate (FDR) and correct for multiple hypothesis testing, a $q$ value was calculated and used to assess the significance of each test (*Benjamini and Hochberg, 1995*). A probe-by-probe comparison was performed between different treatments using water treatment as the reference sample. In each comparison, a $q$ value and fold change (FC) were computed for each gene locus. The gene expression fold changes were computed based on the normalized log-transformed signal intensity data. The comparison results were further explored to obtain numbers of overlapped genes between $NAD^+$ treatment and *Pst* DC3000/*avrRpt2* infection. Pathway and gene ontology analysis were performed on deferentially expressed genes using DAVID Bioinformatics Resources (*Huang et al., 2009*).

## Microscopy

N.*N. benthamiana* leaf tissues were mounted in water and viewed with a Zeiss confocal LSM 5 Pascal microscope (Jena, Germany). GFP was visualized using an excitation wavelength of 488 nm and a bandpass 505 to 530 nm emission filter and mCherry was visualized using an excitation of 543 nm and a bandpass 600 to 680 nm emission filter.

## Statistical methods

Statistical analyses were performed using the one-way ANOVA and the two-way ANOVA in Prism 5.0b (GraphPad Software, La Jolla, CA).

## Accession numbers

The accession numbers of the microarrays discussed in this manuscript are GSE76568 and GSE38986 in Gene Expression Omnibus.

## Acknowledgements

We thank Dr. Jeffrey A Rollins (University of Florida) for critical comments on the manuscript, Dr. Heather McAuslane for helping with the cabbage looper eggs, the Arabidopsis Biological Resource Center at The Ohio State University (Columbus, OH) for the Salk and Sail lines, and the European Arabidopsis Stock Centre at The University of Nottingham (Nottingham, UK) for the Gabi line. This work was partially supported by a grant from the National Science Foundation (IOS-0842716) awarded to ZM. Publication of this article was funded in part by the University of Florida Open Access Publishing Fund.

## Additional information

### Funding

| Funder | Grant reference number | Author |
|---|---|---|
| National Science Foundation | IOS-0842716 | Zhonglin Mou |

The funders had no role in study design, data collection and interpretation, or the decision to submit the work for publication.

### Author contributions

CW, MZ, Conceptualization, Formal analysis, Investigation, Writing—original draft, Writing—review and editing; XZ, Formal analysis, Investigation, Project administration; JY, Data curation, Formal analysis, Methodology, Writing—original draft; YZ, Formal analysis, Investigation, Methodology; ZM, Conceptualization, Funding acquisition, Writing—original draft, Writing—review and editing

### Author ORCIDs

Mingqi Zhou, http://orcid.org/0000-0002-4605-5467
Zhonglin Mou, http://orcid.org/0000-0003-0243-4905

## Additional files

### Supplementary files

• Supplementary file 1. (A) Well-known defense genes affected by $NAD^+$ treatment. (B) Receptor-like genes induced by $NAD^+$ treatment. (C) Receptor-like kinase genes induced by *P. brassicae* oviposition and/or $NAD^+$ treatment. (D) Primers used in this study.

### Major datasets

The following dataset was generated:

| | Database, license, and accessibility |
|---|---|

| Author(s) | Year | Dataset title | Dataset URL | information |
|---|---|---|---|---|
| Wang C, Zhou M, Zhang X, Yao J, Zhang Y, Mou Z | 2016 | Microarray analysis of exogenous NAD+-induced transcriptome changes in Arabidopsis thaliana | https://www.ncbi.nlm.nih.gov/geo/query/acc.cgi?acc=GSE76568 | Publicly available at the NCBI Gene Expression Omnibus (accession no: GSE76568) |

The following previously published dataset was used:

| Author(s) | Year | Dataset title | Dataset URL | Database, license, and accessibility information |
|---|---|---|---|---|
| Wang Y, Zhang Y, Yao J, Sun Y, Mou Z | 2012 | Microarray analysis of Atelp2, npr1 and wild type (Col-0) infected with the avirulent bacterial pathogen Pst DC3000/avrRpt2 | https://www.ncbi.nlm.nih.gov/geo/query/acc.cgi?acc=GSE38986 | Publicly available at the NCBI Gene Expression Omnibus (accession no: GSE38986) |

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
