## [Decision Letter]

Thank you for submitting your article "A lectin receptor kinase senses extracellular nicotinamide adenine dinucleotide in *Arabidopsis thaliana*" for consideration by *eLife*. Your article has been favorably evaluated by Detlef Weigel (Senior Editor) and three reviewers, one of whom is a member of our Board of Reviewing Editors. The reviewers have opted to remain anonymous.

The reviewers have discussed the reviews with one another and with the Senior Editor, and the Reviewing Editor has summarized essential revisions below.

The referees greatly appreciated the identification of Arabidopsis LecRK-I.8 as a receptor like protein that mediates bacterial immunity in plants and that is implicated in host-derived damage-associated molecular pattern recognition, clearly a timely and important topic. The genetic evidence provided in support of this conclusion is very convincing. Questions remain, however, as to whether this protein is indeed a bona fide receptor for eNAD+. Essential revisions include the following:

1) Immunoprecipitation of epitope-tagged LecRK-I.8 is likely to pull-down numerous other proteins closely associated with the bait in membrane micro domains and/or supramolecular protein assemblies. To strengthen the suite of binding experiments you should either perform microsome-based binding assays in vivo (from WT and lecrk-I.8 mutants) or alternatively, use recombinant ectodomain protein in SPR assays (ITC, MST also appropriate) to conduct appropriate, quantitative biochemical binding assays. Such assays might also reveal the presence of the predicted second, low-affinity receptor.

2) It is also of note that all three mutant genotypes showed substantial residual eNAD+ responses, suggesting functional redundancy of LecRK-I.8 with another, yet unknown protein. Likely, this is LecRK-I.6. Therefore, it would be important to know whether a double mutant would indeed show less or no eNAD+ responsiveness.

3) The reported dissociation constant of the receptor with eNAD+ is approx. 450 nM, which is in stark contrast with much higher eNAD+ concentrations used to trigger plant defenses. Here, a range of much lower eNAD+ concentrations should be used in early PTI assays (ROS burst etc.) to establish clearly a link between the kD and the EC50 value for a given PTI response.

4) Results shown in Figure 3 are insufficient to strongly demonstrate membrane-localization of LecRK-I.8.

5) In Figure 3, both a positive control such as DORN1 and a negative control like GFP are missing. Likewise, how often has this effect been observed and how can the authors discriminate between free GFP and epitope-tagged LecRK-I.8?

[Editors' note: further revisions were requested prior to acceptance, as described below.]

Thank you for resubmitting your work entitled "A lectin receptor kinase senses extracellular nicotinamide adenine dinucleotide in *Arabidopsis thaliana*" for further consideration at *eLife*. Your article has been favorably evaluated by Detlef Weigel (Senior Editor) and three reviewers, one of whom is a member of our Board of Reviewing Editors.

The manuscript has been improved but there are some remaining issues that need to be addressed before acceptance.

All referees acknowledge the improvements during the revision, but they continue to have concerns regarding ligand binding of LecRK-I.8 that you should address before final approval. We do not expect you to conduct additional experiments here, but you should include clear statements that:

i) there are differences in ligand affinities between OE and wild-type plants, andii) the significant differences between Kd values and the concentrations used to trigger defenses are rather unusual and difficult to reconcile with a primary ligand sensor function of LecRK-I.8.

Our sense is that while LecRK-I.8 is surely implicated in eNAD+ perception, your data suggest rather than show that this protein is indeed the bona fide ligand binding site. With such an explanatory statement added we will be happy to accept your manuscript. This should be reflected in title and Abstract. E.g.

"A lectin receptor kinase *as a potential sensor* for extracellular nicotinamide adenine dinucleotide…"

and

"Here, we identify a lectin receptor kinase (LecRK), LecRK-I.8, as a *potential* eNAD+ receptor in Arabidopsis. The extracellular lectin domain of LecRK-I.8 *deleted* binds NAD+ with a dissociation constant of 436.5 ± 104.8 nM, *although much higher concentrations are needed to trigger* in vivo *responses*. Transferred DNA insertion knockout of LecRK-I.8 particularly inhibits NAD+-induced immune responses, whereas overexpression of LecRK-I.8 enhances the Arabidopsis response to NAD+. Furthermore, we show that LecRK-I.8 is required for Arabidopsis basal resistance against bacterial pathogens, substantiating a role for eNAD+ in plant immunity. Our results demonstrate that cell-surface lectin receptors can *potentially* function as eNAD+-binding receptors and provide direct evidence for eNAD+ being a bona fide endogenous signaling molecule in plants."

---

## [Author Response]

*The referees greatly appreciated the identification of Arabidopsis LecRK-I.8 as a receptor like protein that mediates bacterial immunity in plants and that is implicated in host-derived damage-associated molecular pattern recognition, clearly a timely and important topic. The genetic evidence provided in support of this conclusion is very convincing. Questions remain, however, as to whether this protein is indeed a bona fide receptor for eNAD+. Essential revisions include the following:*

*1) Immunoprecipitation of epitope-tagged LecRK-I.8 is likely to pull-down numerous other proteins closely associated with the bait in membrane micro domains and/or supramolecular protein assemblies. To strengthen the suite of binding experiments you should either perform microsome-based binding assays* in vivo *(from WT and lecrk-I.8 mutants) or alternatively, use recombinant ectodomain protein in SPR assays (ITC, MST also appropriate) to conduct appropriate, quantitative biochemical binding assays. Such assays might also reveal the presence of the predicted second, low-affinity receptor.*

As suggested, we performed microsome-based binding assays using wild-type, lecrk-I.8-2, and 35S:LecRK-I.8 plants. We observed a dramatic increase of NAD^+^ binding activity in the membrane fractions of the *35S:LecRK-I.8* transgenic plants and a clear decrease of NAD^+^ binding activity in the membrane fractions of the *lecrk-I.8-2* mutant plants. Furthermore, the lecrk-I.8-2 mutant membrane fractions still showed certain levels of NAD+ binding activity, indicating that there are other NAD+ receptors in Arabidopsis. These results are presented in Figure 5 and in the text.

*2) It is also of note that all three mutant genotypes showed substantial residual eNAD+ responses, suggesting functional redundancy of LecRK-I.8 with another, yet unknown protein. Likely, this is LecRK-I.6. Therefore, it would be important to know whether a double mutant would indeed show less or no eNAD+ responsiveness.*

We thank the editor and the reviewers for this suggestion. We cloned the cDNA fragment encoding the extracellular domain of LecRK-I.6 into pMAL-p2x and purified the MBP-eLecRK-I.6 fusion protein. NAD+ binding assays revealed that while simultaneously purified MBP-eLecRK-I.8 bound NAD+, MBP and MBP-eLecRK-I.6 did not (Figure 5—figure supplement 2), indicating that LecRK-I.6 is not an eNAD+-binding receptor. This result has been added in the text. Although we have transformed a CRISPR/Cas9 construct into lecrk-I.8-2 to create a double mutant, it is not necessary to wait for the double mutant based on the NAD+ binding assay result.

*3) The reported dissociation constant of the receptor with eNAD+ is approx. 450 nM, which is in stark contrast with much higher eNAD+ concentrations used to trigger plant defenses. Here, a range of much lower eNAD+ concentrations should be used in early PTI assays (ROS burst etc.) to establish clearly a link between the kD and the EC50 value for a given PTI response.*

As suggested, we used a series of low concentrations (0, 0.5, 5, and 50 μM) of NAD+ to treat the plants and analyzed the induction of several early PAMP-responsive genes including GST1, FRK1, NHO1, and WRKY29. As shown in Figure 5—figure supplement 3, 5 μM NAD+ significantly induced GST1 and FRK1 expression. This result has been added in the text. Moreover, it seems that 0.5 μM of NAD+ is able to induce FRK1, although the induction was not statistically significant.

*4) Results shown in Figure 3 are insufficient to strongly demonstrate membrane-localization of LecRK-I.8.*

To further demonstrate that LecRK-I.8 is localized at the plasma membrane, we co-transformed LecRK-I.8-GFP and AHA2-mCherry into *N. benthamiana* and found that the GFP and mCherry signals precisely overlapped with each other. Since AHA2 is a well-established plasma membrane localized P-type H^+^-ATPase, this result indicates that LecRK-I.8 is localized at the plasma membrane. The result is presented in Figure 3 and in the text.

*5) In Figure 3, both a positive control such as DORN1 and a negative control like GFP are missing. Likewise, how often has this effect been observed and how can the authors discriminate between free GFP and epitope-tagged LecRK-I.8?*

We thank the editor and the reviewers for this point. We used DORN1-GFP as the positive control and GFP as the negative control. To our surprise, expression of GFP alone also appeared to affect plasmolysis to some extent. We repeated the experiment many times and tested different time points. Unfortunately, we do not feel confident to conclude that expression of GFP alone does not affect plasmolysis. For this reason, we removed the plasmolysis results in the revised manuscript.

We think that the results from the co-localization experiment and the microsomal binding assays are sufficient to demonstrate the plasma membrane localization of LecRK-I.8.

GFP and LecRK-I.8-GFP can be easily distinguished by their subcellular localization. While GFP is localized in the cytoplasm and nucleus, LecRK-I.8-GFP is in the plasma membrane (Figure 3).

[Editors' note: further revisions were requested prior to acceptance, as described below.]

*The manuscript has been improved but there are some remaining issues that need to be addressed before acceptance.*

*All referees acknowledge the improvements during the revision, but they continue to have concerns regarding ligand binding of LecRK-I.8 that you should address before final approval. We do not expect you to conduct additional experiments here, but you should include clear statements that:*

*i) there are differences in ligand affinities between OE and wild-type plants, andii) the significant differences between Kd values and the concentrations used to trigger defenses are rather unusual and difficult to reconcile with a primary ligand sensor function of LecRK-I.8.*

*Our sense is that while LecRK-I.8 is surely implicated in eNAD+ perception, your data suggest rather than show that this protein is indeed the bona fide ligand binding site. With such an explanatory statement added we will be happy to accept your manuscript. This should be reflected in title and Abstract. E.g.*

*"A lectin receptor kinase as a potential sensor for extracellular nicotinamide adenine dinucleotide…"*

*and*

*"Here, we identify a lectin receptor kinase (LecRK), LecRK-I.8, as a potential eNAD+ receptor in Arabidopsis. The extracellular lectin domain of LecRK-I.8 deleted binds NAD+ with a dissociation constant of 436.5 ± 104.8 nM, although much higher concentrations are needed to trigger* in vivo *responses. Transferred DNA insertion knockout of LecRK-I.8 particularly inhibits NAD+-induced immune responses, whereas overexpression of LecRK-I.8 enhances the Arabidopsis response to NAD+. Furthermore, we show that LecRK-I.8 is required for Arabidopsis basal resistance against bacterial pathogens, substantiating a role for eNAD+ in plant immunity. Our results demonstrate that cell-surface lectin receptors can *potentially* function as eNAD+-binding receptors and provide direct evidence for eNAD+ being a bona fide endogenous signaling molecule in plants."*

Based on your suggestions, we have made the following changes in the revised manuscript.

1) As suggested, the title has been changed to “A lectin receptor kinase as a potential sensor for extracellular nicotinamide adenine dinucleotide in *Arabidopsis thaliana*”.

2) Changes have also been made in the Abstract as suggested: “potential” was added; “although much higher concentration are needed to trigger in vivo responses” was added; and “specifically” was deleted. Additionally, the Abstract was shortened to 150 words.

3) In the last paragraph of the Introduction, “potential” and “potentially” was added.

4) In the subsection “LecRK-I.8 specifically binds NAD^+^”, first paragraph: “Interestingly, there were differences in ligand affinities between the membrane fractions of the *35S:LecRK-I.8* plants and the wild type, which might suggest possible conformation changes when LecRK-I.8 is overexpressed in plants.” was added; “in line with this result” was replaced by “on the other hand”; “indicating that the NAD^+^ concentrations required to trigger defense responses are significantly higher than the *K*_d_ value of LecRK-I.8, which is rather unusual and difficult to reconcile with a primary ligand sensor function of LecRK-I.8.” was added.

5) In the last paragraph of the subsection “*LecRK-I.8* specifically binds NAD^+^”: “demonstrates” was replaced with “suggests” and “is” was replaced with “may be”. At the end of the subsection “Mutations in *LecRK-I.8* particularly inhibit NAD^+^-induced defense responses”, “support the conclusion” and “specific” were replaced with “suggest” and potential”, respectively.

6) In the first and second paragraphs of the Discussion, “potential” was added; “bona fide” was replaced with “potential”; and “demonstrates” was replaced with “suggest”.

7) In the third paragraph of the Discussion, “potential” was added.

8) In the fourth paragraph of the Discussion, “potentially” was added.